# Prognostic Significance of COX-2 Overexpression in *BRAF*-Mutated Middle Eastern Papillary Thyroid Carcinoma

**DOI:** 10.3390/ijms21249498

**Published:** 2020-12-14

**Authors:** Sandeep Kumar Parvathareddy, Abdul K. Siraj, Padmanaban Annaiyappanaidu, Saif S. Al-Sobhi, Fouad Al-Dayel, Khawla S. Al-Kuraya

**Affiliations:** 1Human Cancer Genomic Research, Research Center, King Faisal Specialist Hospital and Research Center, P.O. Box 3354, Riyadh 11211, Saudi Arabia; psandeepkumar@kfshrc.edu.sa (S.K.P.); asiraj@kfshrc.edu.sa (A.K.S.); pannaiyappanaidu97@kfshrc.edu.sa (P.A.); 2Department of Surgery, King Faisal Specialist Hospital and Research Center, P.O. Box 3354, Riyadh 11211, Saudi Arabia; sobhi@kfshrc.edu.sa; 3Department of Pathology, King Faisal Specialist Hospital and Research Centre, P.O. Box 3354, Riyadh 11211, Saudi Arabia; dayelf@kfshrc.edu.sa

**Keywords:** cyclooxygenase-2, *BRAF* mutation, papillary thyroid carcinoma, disease-free survival

## Abstract

The cyclooxygenase-2 (COX-2)–prostaglandin E2 (PGE2) pathway has been implicated in carcinogenesis, with *BRAF* mutation shown to promote PGE2 synthesis. This study was conducted to evaluate COX-2 expression in a large cohort of Middle Eastern papillary thyroid carcinoma (PTC), and further evaluate the prognostic significance of COX-2 expression in strata of *BRAF* mutation status. *BRAF* mutation analysis was performed using Sanger sequencing, and COX-2 expression was evaluated immunohistochemically using tissue microarray (TMA). COX-2 overexpression, noted in 43.2% (567/1314) of cases, was significantly associated with poor prognostic markers such as extra-thyroidal extension, lymph-node metastasis, and higher tumor stage. COX-2 was also an independent predictor of poor disease-free survival (DFS). Most notably, the association of COX-2 expression with DFS differed by *BRAF* mutation status. COX-2 overexpression was associated with poor DFS in *BRAF*-mutant but not *BRAF* wild-type PTCs, with a multivariate-adjusted hazard ratio of 2.10 (95% CI = 1.52–2.92; *p* < 0.0001) for COX-2 overexpressed tumors in *BRAF*-mutant PTC. In conclusion, the current study shows that COX-2 plays a key role in prognosis of PTC patients, especially in *BRAF-*mutated tumors. Our data suggest the potential therapeutic role of COX-2 inhibition in patients with *BRAF*-mutated PTC.

## 1. Introduction

Thyroid cancer is the most common endocrine malignancy. Its incidence is steadily rising worldwide [1,2], with the most common histology being papillary thyroid carcinoma (PTC) [3]. The incidence of PTC in Saudi Arabia is high—it is the second most common cancer affecting Saudi women, after breast cancer [4]. Although PTC is an indolent and slow-growing malignancy that can be successfully treated, there are patients that progress to more aggressive disease, with recurrence and metastasis [5,6,7]. It is critical to identify this subset of patients who might benefit from more aggressive therapy. Therefore, identification of a molecular marker that can be useful for prognostication of PTC patients is critically needed.

Tumor-promoting inflammation is a promising target for cancer therapy [8] due to its role in promoting cancer progression by increasing several growth, angiogenic, and immunosuppressive factors [9]. One of the important processes causing cancer inflammation is the cyclooxygenase-2 (COX-2) pathway. COX is a group of enzymes that are required for prostaglandin E2 (PGE2) synthesis [10]. In normal physiological status, COX-2 and PGE2 are upregulated and act as pro-inflammatory factors [11]. Although COX-2 expression is usually undetectable in normal tissue, it has been observed to be overexpressed in several human cancers [12,13,14,15]. Many studies have described the mechanisms by which COX-2 can promote carcinogenesis, including inhibition of apoptosis, increase in cell proliferation, stimulation of angiogenesis, and development of a tumor-promoting inflammatory microenvironment [9,16,17,18,19,20].

COX-2 overexpression in different thyroid cancer histotypes has also been investigated in several studies [21,22,23], the majority of which have been aimed at evaluation of its prognostic relevance by correlating the expression levels with patients’ clinicopathological features. Other studies have attempted to estimate the prognostic value of COX-2 expression in thyroid cancer [15,24]. Those studies found that COX-2 overexpression in thyroid cancer is associated with aggressive clinical behavior and tumor recurrence [15,25].

Another characteristic of PTC is that a single nucleotide mutation in the *BRAF* gene, V600E, is detected in a majority of PTC patients [26,27], often associated with aggressive disease [7,28]. *BRAFV600E* mutations have been shown to induce oncogenic cellular proliferation by constitutively activating the mitogen-activated protein (MAP) kinase pathway [29]. A recent study by Kosumin et al. [30] showed the interactive role of *BRAF* mutation status and COX-2 overexpression in the prognostication of colorectal cancer patients, suggesting that upregulation ofMAP kinase pathway mediates overexpression of COX-2 in *BRAFV600E* tumor cells.

Therefore, we conducted this study to evaluate COX-2 overexpression in a large cohort (n = 1335) of Middle Eastern PTC, and to analyze its correlation with clinicopathological parameters. Then we further evaluated the prognostic significance of COX-2 expression in strata of *BRAF* mutation status.

## 2. Results

### 2.1. Patient Characteristics

The mean age of the study population was 40.4 years (SD = ±16.1 years), with a male-to-female ratio of 1:3. A majority of the cases were of classical variant (67.1%) and Stage I tumors (83.4%). A nearly equal proportion of tumors exhibited presence and absence of extra-thyroidal extension, as well as multifocality (Table 1). Patient characteristics stratified by *BRAF* mutation status is presented in Table 1.

### 2.2. Clinicopathological Associations of COX-2 Expression in PTC

We performed immunohistochemical analysis to look for COX-2 protein expression in our cohort of 1335 PTC cases. Immunohistochemistry data were interpretable in 1314 cases, and cytoplasmic staining was noted (Figure 1). COX-2 overexpression, noted in 43.2% (567/1314) of cases, was significantly associated with older age (*p* < 0.0001), extra-thyroidal extension (*p* < 0.0001), T4 tumors (*p* = 0.0094), lymph-node metastasis (*p* = 0.0003), and advanced stage (*p* < 0.0001) (Table 2). A significant association was also observed between COX-2 overexpression and poor disease-free survival (DFS; *p* < 0.0001) (Table 2; Figure 2A). Multivariate analysis using the Cox regression model, after adjusting for possible confounding factors of survival, showed that COX-2 overexpression was an independent predictor of poor DFS (HR = 1.57; 95% CI = 1.22–2.03; *p* = 0.0004) (Table 3).

### 2.3. COX-2 Expression and BRAF Mutation in PTC

We found a significant association between COX-2 overexpression and *BRAF* mutation (*p* = 0.0055) in our cohort. Next, we examined the prognostic association of COX-2 expression in the strata of *BRAF* mutation status. Using Kaplan–Meier survival analyses, we found COX-2 expression was associated with a significantly shorter DFS in *BRAF*-mutant PTC, but not in *BRAF* wild-type cases (Figure 2B,C). Multivariate analysis showed COX-2 overexpression was associated with shorter DFS in *BRAF*-mutant PTC cases (HR = 2.10; 95% CI = 1.52–2.92; *p* < 0.0001) (Table 4).

## 3. Discussion

We conducted this study using a large cohort of more than 1300 Middle Eastern PTC tumors, and found a frequency of COX-2 expression of 43.2%. There was a strong association of COX-2 expression with adverse markers of PTC prognosis, most notably extra-thyroidal extension, lymph-node metastasis, and higher tumor stage. A recent study by Fu et al. [15] demonstrated a similar association of COX-2 expression with aggressive clinicopathological features such as extra-thyroidal extension and multifocality in 252 PTC samples. Several previous studies have explored the usefulness of COX-2 as a biomarker of thyroid malignancies and its potential role in PTC carcinogenesis [15,31,32,33], but the prognostic role of COX-2 in PTC is still controversial [34,35]. Our current study shows that elevated COX-2 expression is an independent predictor of poor DFS in patients with PTC.

COX-2 mainly generates prostaglandin E2 (PGE2). The COX-2–PGE2 pathway is known to play an important role in tumor progression, and its oncogenic role has been shown in several tumor types [36,37,38,39]. However, recent evidence shows that PGE2 is important in modifying the tumor microenvironment to induce immune tolerance [40,41,42]. PGE2 induces alterations in cytokine balance and causes suppression of lymphocyte proliferation following mitogen stimulation and inhibition of dendritic cells [43,44,45]. Given the known function of PGE2 in modulating the tumor microenvironment to suppress an immune response [46], our data clearly show the strong relationship among cancer progression, aggressiveness, lymph-node metastasis, and COX-2 expression.

We observed a differential prognostic association of COX-2 expression according to *BRAF* mutation status in our study. The prognostic association of COX-2 expression was significantly pronounced in *BRAF*-mutated PTC compared to *BRAF* wild-type PTC. A similar observation has been documented recently in CRC [30]. This further supports the role of *BRAF* mutation in the acceleration of the production of PGE2 via COX-2 [41,47,48], and highlights how a subset of PTC might use increased COX-2 activity as a possible pathway that affects the survival of patients with the *BRAF* mutation.

Our study reveals the necessity for additional prognostic biomarkers for PTC, especially markers of tumor-associated inflammation, which could be used to tailor therapeutic approaches and improve patient survival. Although our findings suggest that inhibition of COX-2 in *BRAF*-mutated PTC might be a good therapeutic strategy, clinical trials and further clinical research is needed to explore the therapeutic advantage of adding COX-2 inhibitors to the current iodine therapy in *BRAF*-mutated PTC patients. The findings of our study are summarized in Figure 3.

## 4. Materials and Methods 

### 4.1. Sample Selection

The study included 1335 PTC patients diagnosed between 1989 and 2018 at King Faisal Specialist Hospital and Research Center (Riyadh, Saudi Arabia) with available archival tissue samples. Clinicopathological data were collected from case records (Table 1). The hospital’s institutional review board approved the collection of archival samples. For this study, since only archived paraffin tissue blocks were used, a waiver of consent was obtained from the Research Advisory Council (RAC) under project RAC# 2110 031.

### 4.2. DNA Isolation

DNA was extracted from PTC formalin-fixed and paraffin-embedded (FFPE) tumor tissues using the Gentra DNA isolation kit (Gentra, Minneapolis, MN, USA) according to the manufacturer’s protocols as elaborated in previous studies [49].

### 4.3. Sanger Sequencing Analysis

The sequencing of entire coding and splicing regions of exon 15 in *BRAF* genes, and exon 2 and 3 in *HRAS* and *NRAS* genes, in the 1335 PTC samples was carried out using Sanger sequencing technology. Primer 3 online software was used to design the primers (available upon request). PCR and Sanger sequencing analysis were conducted as described previously [50]. The reference sequence was downloaded from NCBI GenBank. The sequencing results were compared with the reference sequence by Mutation Surveyor V4.04 (Soft Genetics, LLC, State College, PA, USA).

### 4.4. Tissue Microarray (TMA) Construction and Immunohistochemistry (IHC) Analysis

The tissue microarray (TMA) format was used for immunohistochemical analysis of the PTC samples. TMA was constructed as previously described [51]. A modified semiautomatic robotic precision instrument (Beecher Instruments, Woodland, WI, USA) was used to punch tissue cylinders with a diameter of 0.6 mm from a representative tumor area of the donor tissue block into the recipient paraffin block. Two 0.6 mm cores of PTC were arrayed from each case.

Tissue microarray slides were processed and stained manually as described previously [52], using a primary antibody against COX-2 (ab15191, 1:1000 dilution, pH 6.0, Abcam, Cambridge, UK). The Dako Envision Plus System kit was used as the secondary detection system with 3,30-diaminobenzidine as chromogen. All slides were counterstained with hematoxylin, dehydrated, cleared, and mounted. The negative controls included omission of the primary antibody. Normal tissues of different organ system were also included in the TMA to serve as controls. Only fresh-cut slides were stained simultaneously to minimize the influence of slide aging and maximize reproducibility of the experiment.

A cytoplasmic staining was observed and scored. COX-2 was scored as described previously [53] using the H score. The intensity of staining was scored from 0–3 (0: absent, 1+: weak, 2+: moderate, 3+: strong), and the proportion of tumor-cell staining for a particular intensity was recorded in 5% increments in the range of 0–100. A final H score was assigned using the following formula: H score = (1 × (% cells 1+) + 2 × (% cells 2+) + 3 × (% cells 3+)). This H score ranged from 0–300. Two scores per tumor were analyzed to minimize the number of missing/uninterpretable spots. However, the higher of the two scores was used as the final score. X-tile plots were constructed for the assessment of biomarkers and the optimization of cut-off points based on an outcome described earlier [54]. Based on X-tile plots, PTC cases were classified into two subgroups: those with an H score of 0, defined as negative expression of COX-2, and those with an H score > 0, defined as overexpression.

### 4.5. Statistical Analysis

The associations between clinico-pathological variables and protein expression were derived using contingency table analysis and Chi square tests. The Mantel–Cox log-rank test was used to evaluate disease-free survival. Survival curves were generated using the Kaplan–Meier method. The Cox proportional-hazards regression model was used for multivariate analysis. Two-sided tests were used for statistical analyses, with a limit of significance defined as *p* value < 0.05. Data analyses were performed using the JMP11.0 (SAS Institute, Inc., Cary, NC, USA) software package.

## 5. Conclusions

Recently, the possibility of combining different therapies, including biological therapies and kinase inhibitors, has contributed to tailoring the treatment of cancer patients and improving their survival. Our data show that COX-2 correlates with PTC disease-free survival in *BRAF*-mutated tumors, representing a useful prognostic marker for risk stratification of thyroid cancer patients. These findings have clinical relevance because they provide a rationale to test COX-2 inhibition as a potential treatment to prevent PTC progression and enhance the antitumor activity of other cancer therapies to treat patients with aggressive PTC and *BRAF* mutations.

## Figures and Tables

**Figure 1 ijms-21-09498-f001:**
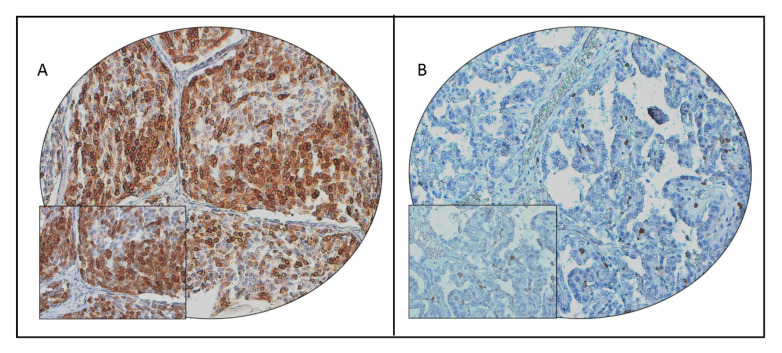
Cyclooxygenase-2 (COX-2) immunohistochemical staining in papillary thyroid carcinoma (PTC) tissue microarray (TMA). Representative examples of tumors showing (**A**) high expression and (**B**) low expression of COX-2 (20×/0.70 objective on an Olympus BX 51 microscope (Olympus America Inc., Center Valley, PA, USA), with the inset showing a 40×/0.85 aperture magnified view of the same TMA spot).

**Figure 2 ijms-21-09498-f002:**
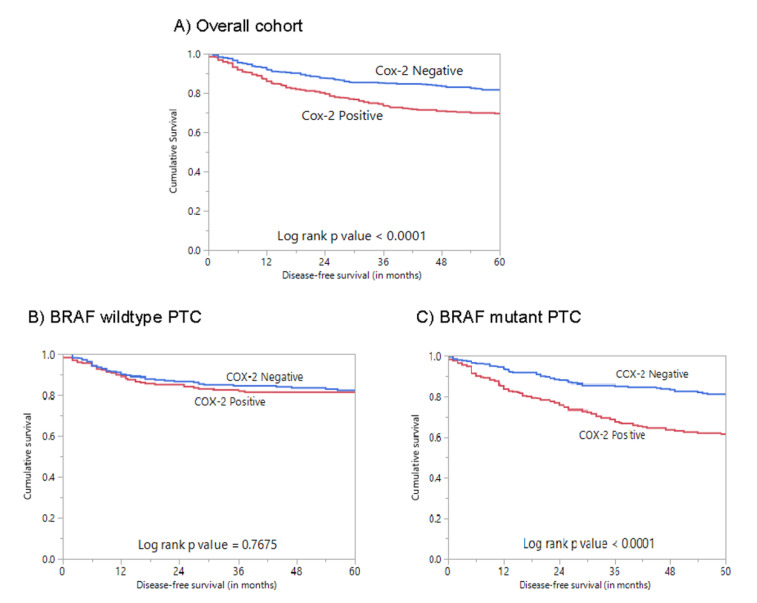
Survival analyses of COX-2 protein expression in the overall cohort and based on *BRAF* mutation status. (**A**) Kaplan–Meier survival plot showing statistically significant poor disease-free survival in COX-2 high-expression cases compared to COX-2 low-expression cases (*p* < 0.0001); (**B**) Kaplan–Meier survival plot showing no statistically significant difference in disease-free survival for COX-2 expression in *BRAF* wild-type PTC (*p* = 0.7675); (**C**) Kaplan–Meier survival plot showing statistically significant poor disease-free survival for cases with COX-2 overexpression in *BRAF*-mutant PTC (*p* < 0.0001).

**Figure 3 ijms-21-09498-f003:**
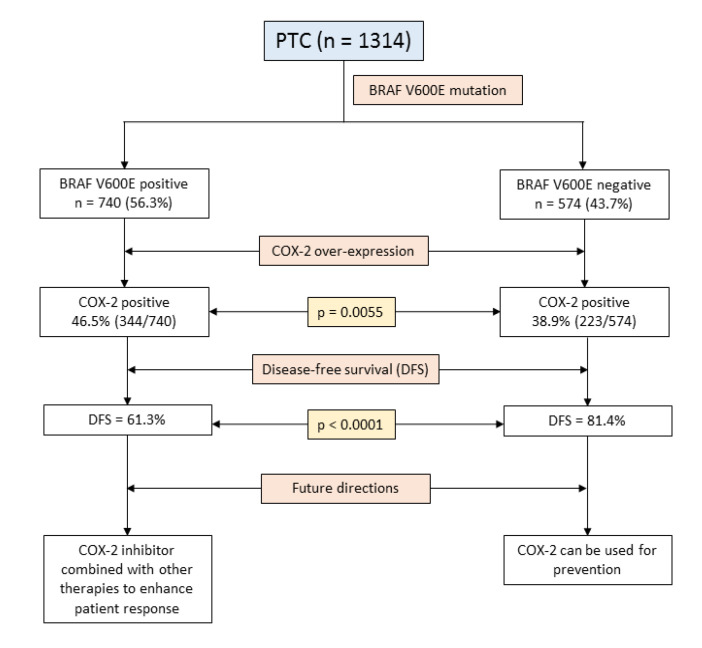
Schematic presentation of the study findings and future directions.

**Table 1 ijms-21-09498-t001:** Patient characteristics according to *BRAF* mutation status.

	Total		*BRAF* Wild-Type		*BRAF* Mutant	
	No.	%	No.	%	No.	%
**Total**	1335		584	43.7	751	56.3
**Age at surgery (years)**						
Mean ± SD	40.4 ± 16.1		36.6 ± 15.5		43.3 ± 15.9	
<55	1087	81.4	509	87.2	578	77.0
≥55	248	18.6	75	12.8	173	23.0
**Sex**						
Male	330	24.7	138	23.6	192	25.6
Female	1005	75.3	446	76.4	559	74.4
**Histologic subtype**						
Classical variant	895	67.1	314	53.7	581	77.4
Follicular variant	232	17.4	185	31.7	47	6.3
Tall cell variant	118	8.8	21	3.6	97	12.9
Other variants	90	6.7	64	11.0	26	3.5
**Extrathyroidal extension**						
Present	590	44.2	178	30.5	412	54.9
Absent	745	55.8	406	69.5	339	45.1
**Multifocality**						
Yes	670	50.2	259	44.4	411	54.7
No	665	49.8	325	55.6	340	45.3
**pT**						
T1	369	28.5	176	31.3	193	26.4
T2	266	20.6	138	24.6	128	17.5
T3	549	42.4	212	37.7	337	46.0
T4	110	8.5	36	6.4	74	10.1
**pN**						
N0	527	39.5	280	48.0	247	32.9
N1	683	51.1	245	41.9	438	58.3
Nx	125	9.4	59	10.1	66	8.8
**pM**						
M0	1205	95.4	518	93.7	687	96.8
M1	58	4.6	35	6.3	23	3.2
**TNM Stage**						
I	1084	83.4	499	87.5	585	80.1
II	148	11.4	51	8.9	97	13.3
III	21	1.6	4	0.7	17	2.3
IV-A	17	1.3	2	0.4	15	2.1
IV-B	30	2.3	14	2.5	16	2.2
***NRAS* mutation**						
Present	85	6.4	83	14.3	2	0.3
Absent	1244	93.6	498	85.7	746	99.7
***HRAS* mutation**						
Present	30	2.3	28	4.2	2	0.3
Absent	1298	97.7	553	95.2	745	99.7

**Table 2 ijms-21-09498-t002:** Association of clinicopathological characteristics with Cox-2 expression in PTC.

	Total		Cox-2 Positive		Cox-2 Negative		*p* Value
	No.	%	No.	%	No.	%	
**Total**	1314		567	43.2	747	56.8	
**Age at surgery (years)**							
<55	1072	81.6	418	39.0	654	61.0	<0.0001
≥55	242	18.4	149	61.6	93	38.4	
**Sex**							
Male	320	24.4	146	45.6	174	54.4	0.3049
Female	994	75.6	421	42.4	174	57.6	
**Histologic subtype**							
Classical variant	879	66.9	394	44.8	485	55.2	0.1542
Follicular variant	229	17.4	84	36.7	145	63.3	
Tall cell variant	116	8.8	52	44.8	64	55.2	
Other variants	90	6.9	37	41.1	53	58.9	
**Extrathyroidal extension**							
Present	579	44.1	291	50.3	288	49.7	<0.0001
Absent	735	55.9	276	37.6	459	62.4	
**Multifocality**							
Yes	658	50.1	295	44.8	363	55.2	0.2175
No	656	49.9	272	41.5	384	58.5	
**pT**							
T1	365	28.7	140	38.4	225	61.6	0.0094
T2	265	20.8	101	38.1	164	61.9	
T3	538	42.2	253	47.0	285	53.0	
T4	106	8.3	53	50.0	53	50.0	
**pN**							
N0	516	43.3	194	37.6	322	62.4	0.0003
N1	675	56.7	325	48.2	350	51.8	
**pM**							
M0	1186	95.4	509	42.9	677	57.1	0.3575
M1	57	4.6	28	49.1	29	50.9	
**TNM Stage**							
I	1069	83.4	415	38.8	654	61.2	<0.0001
II	146	11.4	86	58.9	60	41.1	
III	20	1.6	17	85.0	3	15.0	
IV-A	17	1.3	10	58.8	7	41.2	
IV-B	29	2.3	21	72.4	8	27.6	
***BRAF* mutation**							
Present	740	56.3	344	46.5	396	53.5	0.0055
Absent	574	43.7	223	38.9	351	61.1	
**5-year disease-free survival**				69.5		81.7	<0.0001

**Table 3 ijms-21-09498-t003:** Univariate and multivariate analysis of COX-2 expression using the Cox proportional-hazards model.

Clinical Parameters	Univariate		Multivariate	
	Hazard Ratio (95% CI)	*p*-Value	Hazard Ratio (95% CI)	*p*-Value
Age				
≥55 years (vs. <55 years)	2.82 (2.21–3.57)	<0.0001	2.04 (1.51–2.76)	<0.0001
Sex				
Male (vs. Female)	0.59 (0.47–0.74)	<0.0001	0.66 (0.51–0.85)	0.0016
Extrathyroidal extension				
Present (vs. Absent)	2.89 (2.24–3.77)	<0.0001	2.26 (1.72–2.99)	<0.0001
pM				
M1 (vs. M0)	4.52 (3.12–6.34)	<0.0001	2.84 (1.74–4.63)	<0.0001
Stage				
IV (vs. I–III)	4.40 (2.84–6.52)	<0.0001	0.83 (0.45–1.54)	0.5560
COX-2 IHC				
Overexpression (vs. low expression)	1.66 (1.32–2.08)	<0.0001	1.57 (1.22–2.03)	0.0004

**Table 4 ijms-21-09498-t004:** Univariate and multivariate analysis of COX-2 expression in *BRAF*-mutant PTC using the Cox proportional-hazards model.

Clinical Parameters	Univariate		Multivariate	
	Risk Ratio (95% CI)	*p*-Value	Risk Ratio (95% CI)	*p*-Value
Age				
≥55 years (vs. <55 years)	2.45 (1.84–3.25)	<0.0001	1.75 (1.22–2.48)	0.0020
Sex				
Male (vs. Female)	0.55 (0.42–0.74)	<0.0001	0.52 (0.39–0.72)	<0.0001
Extrathyroidal extension				
Present (vs. Absent)	2.77 (1.96–4.01)	<0.0001	2.11 (1.46–3.10)	<0.0001
pM				
M1 (vs. M0)	4.69 (2.64–7.72)	<0.0001	4.44 (1.61–11.71)	0.0036
Stage				
IV (vs. I–III)	3.15 (1.77–5.16)	<0.0001	0.49 (0.17–1.25)	0.1656
COX-2 IHC				
Overexpression (vs. low expression)	2.10 (1.59–2.79)	<0.0001	2.10 (1.52–2.92)	<0.0001

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
