# Peer review of "Prognostic Significance of COX-2 Overexpression in BRAF-Mutated Middle Eastern Papillary Thyroid Carcinoma"

_ijms, 2020, doi:10.3390/ijms21249498_

Round 1

Reviewer 1 Report

This is an interesting study to demonstrate the significance of COX-2 overexpression in BRAF mutated PTC. The study was well designed and performed in a systematic way.

However, some suggestions to improve this manuscript are:

  1. A schematic diagram can be included to summarize the findings and include future directions.
  2. Introduction section is too short, can be expanded.
  3. Conclusion section should be elaborated.
  4. Figure 2 seems to be blurred, can improve the resolution/quality of image.

Author Response

This is an interesting study to demonstrate the significance of COX-2 overexpression in BRAF mutated PTC. The study was well designed and performed in a systematic way.

We thank the reviewer for taking the time to review our manuscript and providing constructive feedback to improve the manuscript. It gives us immense pleasure to know that the reviewer finds our study to be “well designed and performed in a systematic way”. The concerns of the reviewer are addressed point by point below.

However, some suggestions to improve this manuscript are:

  1. A schematic diagram can be included to summarize the findings and include future directions.

We thank the reviewer for their valuable suggestion. As suggested by the reviewer, we have now included a schematic diagram to summarize the findings and future directions. This has been incorporated in the manuscript as Fig. 3 (Page 8, Line 176).

  1. Introduction section is too short, can be expanded.

We appreciate the reviewer’s concern regarding the Introduction section being too short. Respecting the reviewer’s suggestion, we have now expanded the Introduction. These changes are reflected in the manuscript (Page 2, Line 62-88).

  1. Conclusion section should be elaborated.

We thank the reviewer for their suggestion to elaborate the Conclusion section. As suggested by the reviewer, we have now expanded the Conclusion section in the manuscript (Page 9, Line 232-239).

  1. Figure 2 seems to be blurred, can improve the resolution/quality of image.

We appreciate the reviewer’s concern regarding the resolution/quality of the image in Figure 2. We would like to assure the reviewer that the current image is at 600 dpi resolution in accordance with the journal guidelines. The image probably only appears blurred in the pdf version.

Reviewer 2 Report

BRAF V600E is primarily present in conventional papillary thyroid cancer. It is associated with an aggressive tumor phenotype and higher risk of recurrent and persistent disease in patients with conventional papillary thyroid cancer.

It would be interesting to Know if any mutations other than BRAF V600E have been detected in your patient cohort.

Author Response

We thank the reviewer for taking the time to review our manuscript and providing constructive feedback to improve the manuscript. The concerns of the reviewer are addressed point by point below.

BRAF V600E is primarily present in conventional papillary thyroid cancer. It is associated with an aggressive tumor phenotype and higher risk of recurrent and persistent disease in patients with conventional papillary thyroid cancer.

It would be interesting to know if any mutations other than BRAFV600E have been detected in your patient cohort.

We acknowledge the reviewer’s interest in knowing if any mutations other than BRAFV600E have been detected in our patient cohort. In this cohort of 1335 PTC patients, NRAS mutation data was available for 1329 cases, with mutation noted in 6.4% (85/1329) of cases, whereas HRAS mutation data was available for 1328 cases, with mutation seen in 2.3% (30/1328) of cases. BRAF and RAS mutations were mutually exclusive, except for four cases (two cases had co-existing BRAF and NRAS mutations; two cases had co-existing BRAF and HRAS mutations). We have now added the data for NRAS and HRAS mutations in Table 1 (Page 3, Line 96) as well as in Methods section (Page 9, Line 193-194).

Round 2

Reviewer 1 Report

The manuscript has been modified appropriately and has incorporated the comments and suggestions.